# The Predictability of Stress Coping Strategies and Psychological Capital on the Psychological Well-Being of Autistic Spectrum Children’s Mothers in the Kingdom of Saudi Arabia

**DOI:** 10.3390/bs14121235

**Published:** 2024-12-23

**Authors:** Nawal A. Al Eid, Sami M. Alshehri, Boshra A. Arnout

**Affiliations:** 1Department of Islamic Studies, College of Hummanities and Social Sciences, Princess Nourah bint Abdulrahman University, P.O. Box 84428, Riyadh 11671, Saudi Arabia; naalaed@pnu.edu.sa; 2College of Education, King Khalid University, Abha 62521, Saudi Arabia; smshehrie@kku.edu.sa; 3Department of Psychology, College of Education, King Khalid University, Abha 62521, Saudi Arabia; 4Department of Psychology, College of Arts, Zagazig University, Zagazig 44519, Egypt

**Keywords:** ASD, stress coping, PsyCap, PWB, autistic spectrum children’s mothers

## Abstract

There is an increasing number of studies concerned with the study of children with autism spectrum disorder (ASD). At the same time, there is a lack of interest in studies on their families, especially on mothers who represent the first breadwinner for a child who suffers from a deficit in social communication with others, reflected in their well-being (PWB). This study aimed to reveal the possibility of predicting the PWB of autistic spectrum children’s mothers through the variables of coping strategies and psychological capital (PsyCap). The study used a predictive, descriptive research method to reveal the ability of the variables—coping strategies (problem-solving, avoidance, support, re-evaluation, remorse) and PsyCap (self-efficacy, optimism, hope, and resilience)—in predicting the PWB of autistic spectrum children’s mothers. The study sample consisted of (248) mothers, to whom coping strategies, PsyCap, and PWB measures were applied. The results showed that there were statistically significant differences between working and housewife mothers of autistic spectrum children in solving problems (t = 3.162, *p* < 0.002), avoidance (t = 1.973, *p* < 0.05), positive coping (t = 2.307, *p* < 0.022), self-efficacy (t = 3.667, *p* < 0.000), resilience (t = 3.338, *p* < 0.001), PsyCap (t = 2.866, *p* < 0.005), and PWB (t = 2.549, *p* < 0.011). Meanwhile, there were no statistically significant differences in social support, problem reassessment, remorse, negative coping, optimism, and hope. Also, there were no statistically significant differences due to the number of children in coping strategies, PsyCap, and PWB. The results also showed that there were statistically significant differences at the level of significance (0.05) between mothers whose age was less than 40 years and those 40 years and older in solving problems (t = 2.093, *p* < 0.037) in favor of mothers of the age group 40 years and older (M = 22.00, SD = 1.22), and avoidance (t = 1.987, *p* < 0.048) in favor of mothers under 40 years of age (M = 6.228, SD = 0.464). However, there were no statistically significant differences in social support, problem reassessment, remorse, positive coping, negative coping, self-efficacy, optimism, hope, resilience, the total degree of PsyCap, and well-being due to the variable of the mother’s age. The regression analysis results showed that optimism and problem-solving contributed to (39.90%) of the total change in PWB for mothers of children with autism spectrum. The study’s findings indicate the need to develop the ability of autistic spectrum children’s mothers to solve problems and their PsyCap, which is represented in self-efficacy, optimism, hope, and resilience, to enhance their PWB, which may have a positive impact on their autistic spectrum child.

## 1. Introduction and Theoretical Background

Disability is an issue all societies face, and it results in many social, psychological, health, and economic problems for the disabled, their family, and their community. One of these disabilities is ASD, as it appears at an early stage in a child’s life and continues throughout life. According to the core characteristics of ASD, as identified by the DSM 5 [1], autistic children have challenges with social communication.

A review of the theoretical literature on the prevalence rates of ASD worldwide found that prevalence rates varied by country, gender, assessment tools, socioeconomic status, and continent in which the survey was conducted [2,3,4,5,6]. A meta-analysis [3] of 27 studies on the prevalence of ASD found that the average prevalence rate of ASD was 9.19 per 1000 population. A recent study [5] in Saudi Arabia found that one in every 40 children had the disorder (2.51%). Also, Alyami et al. [4] found that ASD prevalence was higher among males than among females (3:1).

Within the literature on ASD, it is considered one of the most severe childhood problems, as this disorder combines mental and social disability, which negatively affects the normal life of the family, especially when it is accompanied by a deficiency in effective parenting skills, as the parents’ way of thinking is affected, and anger outbursts increase, which affects the child and their behavior skills [7]. Hence, a child’s upbringing with ASD presents unique challenges that may affect the mental health of parents, as well as affect parental involvement and control over social skills, which leads to poor PWB of the entire family [8]. Many studies [8,9] revealed the susceptibility of parents of autistic spectrum children to depression.

The problem of disability is one multifaceted problem, as its impact does not include the disabled child only but the entire family, as children with disabilities and their families suffer from negative attitudes from society, which increases the disabled family’s problems [10]. Also, Imtair and Zulaitny [7] found that the stress resulting from the presence of an autistic spectrum child may cause depression and burnout for the family.

The disability of the autistic spectrum child is a stressful situation that may lead to a change in the roles and expectations of the family members and the accompanying reactions of the parents as a result of their loss of the hopes and aspirations associated with the birth of the child. Discovering symptoms of autism in children represents a shock to the family, causing stressful conditions psychologically, socially, and economically; it is considered the starting point for a series of general stresses for the family, especially the mother [11].

The presence of an autistic child in the family is stressful for the mother [12,13]. The results of previous studies [14] found a high level of psychological stress among mothers of autistic spectrum children. The previous studies [15,16] found that the level of psychological stress among mothers with autism spectrum disorder is higher than the level of psychological stress among fathers. Varghese and Venkatesan [17] revealed that mothers of autistic spectrum children suffer from burnout symptoms, and the results also showed mothers’ need for psychological services, social support, and training on skills to deal with psychological burnout. The results of Al-Mahdi’s study [18] also found that mothers of autistic spectrum children suffer from stress due to the inability of their children to meet the requirements of daily living and the lack of community support.

The autistic spectrum children’s parents are shocked when the diagnosis of their children with autism spectrum is confirmed, followed by denial. Their reactions appear in many symptoms, including approaching other doctors for a second opinion in the hope that the first doctor’s diagnosis is incorrect; then, the parents project their painful thoughts leads to reactions of grief, guilt, and remorse, a stage of going abroad and looking for support, and finally, a reaction of acceptance [19,20,21].

Thus, it becomes clear that different disabilities have negative effects on the child and their family, causing psychological stress [22]. International statistics indicate that 80% of modern diseases are caused by psychological stress that individuals fail to confront effectively, regardless of the type of coping strategies used [23]. Coping strategies with stress are considered factors that help humans maintain psychological and social compatibility in the face of stressful life events and reduce their negative outcomes [8,24]. Coping is the individual’s efforts in dealing with stressful events and is classified into problem-focused and emotion-focused coping. Thus, the strategies for dealing with stress indicate a tendency to respond effectively to stressful events and seek information to solve them [25]. It also represents the behavioral, cognitive, or emotional reactions of mothers of children on the autistic spectrum towards situations and events that cause stress to relieve their psychological state and reduce the impact of the stressful event on mothers [8].

Emotion-focused coping strategies are the efforts to regulate emotions and reduce hardship and emotional distress, such as avoidance, substitution, positive re-evaluation, and control of stress and emotions [26]. By contrast, coping strategies focus on solving the problem by dealing directly with the stressful situation and trying to control it directly by using one’s resources to solve the problem; through the strategy of searching for a solution to the problem, and on seeking social support, appreciation, and care through emotional, material, or moral support from others and cultural environment [27].

Abdullahi [28] showed that parents of autistic spectrum children use coping strategies through personal resources, support from family and friends, community resources, and religious strategies to cope with the stress resulting from their negative thoughts that they cannot change the behavior of their autistic child. A child with an ASD in essential communication and social interaction skills generates inappropriate interaction patterns with their parents, which increases parents’ stress and discomfort leading to increased inappropriate interactions between the child and his parents, so there is a need to train parents to reduce their stress [8,19,28].

Mostafa’s study [29] found that parents of autistic spectrum children used coping strategies for stress resulting from their child’s disorder, such as searching for information and knowledge, avoidance, and denying that information; this results recommended the necessity of providing support to the parents of autistic spectrum children.

Many factors affect coping strategies, including personal variables such as locus of control, self-esteem, personality type, psychological hardness, and self-efficacy, as well as demographic variables such as chronological age, gender, and economic level [30].

Formal and informal social support for mothers of children with autism spectrum is considered an essential source of stress coping resulting from the disability, as it increases the ability of the parents to assume the responsibility of caring for the child as well as their ability to face the stress associated with the problems of the disabled child in general and the autistic child in particular [19]. Abdel-Aziz’s study [31] showed that mothers of autistic spectrum children have several needs for support, including social support, followed by cognitive and training support, and then financial support. Miranda et al. [32] indicated the need for mothers of autistic spectrum children for social support to cope with the stress resulting from their children’s disabilities.

Many studies [33,34,35] indicated that families with autistic spectrum children need education, psychological care, daily care, and support programs for their children, in addition to financial, emotional, and cognitive mental health support. Previous studies have also shown that there is a statistically significant correlation between social support and services provided to children with autism spectrum and the quality of life of their families [36,37,38]. Thus, social support is considered one of the most important resources for improving the quality of family life [39]. Brown et al. [40] and Obst et al. [41] indicated that the quality of support provided to families with disabilities, rather than its quantity, is the strongest indicator of the quality of life of families.

Ashkib and Awena [42] found that the level of psychological stress among mothers of children with ASD is a result of lack of social support. Smith et al. [43] revealed that there is a significant correlation between social support and PWB for mothers of children with autism spectrum. Social support had a statistically significant effect on the mental health of mothers of adolescents and adults with autism spectrum. In the same context, Davis et al. [44] found that one of the strongest predictors of the quality of life of families with disabilities is support from extended family members. These studies indicated the role that social support plays in the ability of mothers of autistic spectrum children to cope with the stress associated with raising their autistic spectrum child.

PsyCap is an important personal resource that enhances individuals’ ability to develop and remain positive in the worst life scenarios; it includes four dimensions: self-efficacy, hope, resilience, and optimism [45,46]. PsyCap is thus generated by individuals’ belief that they can pursue their goals, persevere in achieving these goals, and mobilize efforts and activities in this regard; the ability to withstand difficult situations and the flexibility to regain balance after experiencing them; and a positive attitude regarding achieving success in the present and future [47]. Self-efficacy is defined as an individual’s confidence in their ability to mobilize the motivation, cognitive resources, and courses of action needed to achieve certain goals; individuals with high self-efficacy tend to generalize the expectations of their ability to perform tasks across new situations and are more positive when faced with difficult and unknown situations [48]. Resilience is the ability to recover from adversity, uncertainty, and failure and adapt to changing and stressful life demands [49]. A statistically significant association has been found between high resilience and good physical and mental health [50]. Optimism refers to an individual’s expectation of achieving positive outcomes [51]. Numerous studies and research have shown the beneficial aspects of optimism in various areas of life, such as physical health, prevention of depression, effective decision-making, and life satisfaction [52].

Rather than just this simple set of the four components mentioned above, PsyCap acts as a higher-level construct of the four positive psychological resources (self-efficacy, optimism, hope, resilience, and predictability of work outcomes) beyond its four components, as PsyCap is an integrated, flexible construct, and thus can be improved [50,52,53].

Given that PsyCap is the best positive construct worth studying for its impact on psychological well-being, research has come to explore the benefits of PsyCap on the psychological well-being of mothers of children with autism spectrum disorder, as it enhances individuals’ ability to manage stressful situations, which increases the quality of life and psychological well-being [51,53]. Because the concept of PsyCap is relatively new, very few studies have examined it in parents of children with autism spectrum disorder. Hence, the authors of the current study hypothesized that PsyCap may contribute to the psychological well-being of mothers with autism spectrum disorder.

PWB is the desired goal that a person seeks to reach, the achievement of which leads to an individual’s feeling contentment, joy, optimism, and self-realization, enabling them to manage themselves, as this affects a person’s view of themself, regardless of stress [54]. The sense of the quality of life and PWB of parents of children with disabilities is not the same as that of parents of non-disabled children because the presence of a child with a disability in the family may be a cause of dissatisfaction and a low sense of the quality of life and affects their willingness to face the resulting stress and the extent of the availability of support from the community and institutions. Zuna et al. [55] found that the quality of life of each member of the family, including the individual with disabilities, affects each other, so it is necessary to consider the quality of life of each family member.

Previous studies have indicated a lower level of quality of life for mothers of autistic spectrum children from a lower educational level [56]; as indicated by the study of Piovesan et al. [57], there was a significant correlation between depressive symptoms in mothers of an autistic spectrum child and a decrease in their quality of life.

Among the factors that may help in facing stress resulting from the challenges of giving birth to an autistic child is the PsyCap of the mother, which includes motivation, hope, optimism, psychological resilience, self-confidence, and self-entitlement, all of which lead to a feeling of happiness and resistance to stress and thus feelings of PWB [58]. Self-efficacy refers to the positive belief of the individual, resilience is the process of good adaptation in the event of trauma, and hope is a pattern of attribution through which positive attitudes are interpreted in a personal and continuous way that is recognized [59,60].

Previous studies [15,16,17,18,42,55,57,61] found that the mothers of children with autism spectrum have more stress than their fathers, and this stress negatively impacts their mental health outcomes because they are constantly struggling between their family responsibilities and their excessive concern for their spectrum autistic children; it becomes clear that there is a need to investigate the predictors of PWB of mothers of children with autism spectrum, given that they are the primary caregivers for the care of their autistic child.

## 2. Purpose of the Present Study

Given the continuous increase in the prevalence of ASD in the Kingdom of Saudi Arabia [2,4,5] and the findings of previous studies, which indicated that there was asignificant impact of child disabilities on mothers’ mental health and stress [15,16,17,18,55,56], it is important to investigate predictors of PWB among mothers of children with ASD Therefore, the current study sought to examine the predictability of the variables of coping strategies and PsyCap with the PWB of mothers of autistic spectrum children.

## 3. Methods

### Study Design and Sample

The current study relied on a descriptive approach to predict the PWB of autistic spectrum children’s mothers through coping strategies with stress and PsyCap. The sample of the study consisted of (248) mothers who were responsible for taking care of their autistic spectrum children citizens residing in the Kingdom of Saudi Arabia, and their ages ranged between (33 and 51) years (Table 1). The study was applied in the period from 14 August 2023 until 17 May 2024 after obtaining approval from the Institutional Ethics Committee of Princess Nourah bint Abdulrahman University (IRB Log Number: 23-0495); written informed consent was obtained from all participants by checking the box “I agree to participate in this study”. One of the study tools was a questionnaire that collected demographic data such as age, gender, number of children, place of residence, number of children with ASD, and access to support from a specialized autism society.

The criteria for including or excluding the participants in this study included the following: they should be mothers of children with ASD, residing in Saudi Arabia, and responsible for raising and caring for one or more children with autism spectrum disorder. According to these criteria, eight cases were excluded as they refused to participate in the study because they were busy with their work, and four mothers were excluded because they did not have autistic children, but had a child with other disorders such as an intellectual disability and Down syndrome.

## 4. Measures

### 4.1. Stress Coping Strategies Checklist (SCSC)

The SCSC was prepared by Abu-Al-Atta [61] to assess coping strategies for mothers of autistic spectrum children. The checklist consists of (28) items distributed over five strategies: problem-solving, avoidance, social support, positive evaluation, and remorse. It is answered with one of the three options: yes (3), to some extent (2), or no (1).

In the current study, the psychometric properties of SCSC were verified. Internal consistency was verified by calculating the correlation coefficients between the items and the total score of the dimension. Table 2 shows the results:

The results in Table 2 indicate that the correlation coefficients between items of the SCSC with their dimensions were positive and statistically significant (*p* < 0.01), and ranged between (0.719 and 0.941). These results indicate the internal consistency of SCSC. The reliability coefficients of re-testing for the dimensions were (0.76, 0.75, 0.71, 0.72, 0.77), respectively. Also, the reliability coefficients of Cronbach’s alpha for the dimensions of SCSC were (0.847, 804, 0.915, 0.899), respectively.

### 4.2. Psychological Capital Scale (PCS)

By reviewing the theoretical literature and measures of PsyCap, the authors of this study developed a PsyCap scale for mothers of children with autism spectrum disorder. The PCS consists of (19) items, and it is answered using a Likert scale ranging from very much agree (4) to disagree (1). To verify the psychometric properties of the PCS, exploratory factor analysis was applied using the principal components method after verifying the analysis of sample adequacy (KMO Kaiser–Meyer–Olkin Measure), which reached (0.952). The results of the exploratory factor analysis showed that all items of the scale were loading on four factors (self-efficacy, optimism, hope, and resilience), explaining 64.78% of the total variance of PsyCap. The self-efficacy factor included five items, and their loadings were (0.836, 0.869, 0.811, 0.835, 877), optimism contained five items, and their loadings were (0.881, 894, 0.902, 0.888, 0.921), hope consisted of 5 items, and their loadings were (0.829, 893, 0.798, 0.856, 0.904), and the resilience factor contained four items, the loadings of which were (0.879, 0.915, 0.894, 0.906). Also, the Pearson correlation coefficients between items and the total score of the dimension and PCS were calculated. Table 3 shows the results.

The findings in Table 3 indicate that the correlation coefficients between items of the PCS with their dimensions were positive and statistically significant and ranged between (0.776 and 0.926). The Pearson correlation coefficients between items and PCS total score were positive and statistically significant (*p* < 0.01), ranging between (0.708 and 0.837). Also, the results showed that the Pearson correlation coefficients between self-efficacy, optimism, hope, and resilience and PCS total score were positive and statistically significant (*p* < 0.01): (0.697, 707, 0.822, 0.912), respectively. These results revealed the internal consistency of the PCS. Also, the reliability coefficients of Cronbach’s alpha for the dimensions and PCS as a whole were (0.821, 0.847, 0.811, 0.829, 0.844), respectively.

### 4.3. Psychological Well-Being Scale (PWBS)

In this study, the researchers prepared a scale consisting of (10) self-report statements on the PWB of autistic spectrum children’s mothers, according to the literature review. Participants answer by choosing one of five alternatives that apply perfectly (5), apply to some extent (4), apply (3), do not apply (2), and do not apply at all (1). The scale’s exploratory factor analysis was applied using the principal components method, after verifying the analysis of sample adequacy (KMO = 0.937). The results showed all items of PWBS were loading on one general factor (PWB), contributing to 79.750% of the total variance of PWB. The loadings of the items were (0.990, 0.993, 0.984, 0.915, 0.951, 0.959, 0.811, 0.579, 0.669, and 0.925). Also, the Pearson correlation coefficients between the items and PWBS as a whole were calculated, and the results found that all correlation coefficients were (0.762, 0.887, 0.820, 0.789, 0.839, 0.933, 0.897, 0.818, 0.924, 0.785), all of which are positive and statistically significant (*p* < 0.01). Reliability was also verified by calculating Cronbach’s α coefficient, which was (0.908), and the reliability of Spearman–Brown’s spilt-half coefficient was (0.899).

## 5. Results

### 5.1. Differences in Stress Coping Strategies, PsyCap, and PWB

#### 5.1.1. The Differences Between Employed and Not Employed Mothers of Children with Autism Spectrum

An Independent-Samples *t*-test was used to calculate the differences in coping strategies, PsyCap, and PWB among employed and not employed mothers of children with autism spectrum, and the results are shown in Table 4.

From the results in Table 4, we can conclude that there are statistically significant differences in problem-solving (t = 3.162, *p* < 0.002), avoidance (t = 1.973, *p* < 0.05), positive coping (t = 2.307, *p* < 0.022), self-efficacy (t = 3.667, *p* < 0.000), resilience (t = 3.338, *p* < 0.001), PsyCap (t = 2.866, *p* < 0.005), and PWB (t = 2.549, *p* < 0.011), and the differences are in favor of employed mothers of autistic spectrum children; however, there are no statistically significant differences in social support, re-appraisal of the problem, remorse, negative coping, optimism, and hope (Figure 1).

#### 5.1.2. Differences Between Mothers of Autistic Spectrum Children Due to the Number of Children

An Independent-Samples *t*-test was used to calculate the differences in coping strategies, PsyCap, and PWB among mothers of children with autism spectrum due to the number of children, and the results are shown in Table 5.

The results shown in Table 5 clearly show that there are no statistically significant differences in coping strategies, PsyCap, and PWB according to the number of children (Figure 2).

#### 5.1.3. Differences in Coping Strategies, PsyCap, and PWB Due to the Age of the Mother of Autistic Spectrum Children

An Independent-Samples *t*-test was used to calculate the differences in coping strategies, PsyCap, and PWB among mothers of children with autism spectrum due to the mother’s age. The results are shown in Table 6.

It is clear from the results shown in Table 6 that there are statistically significant differences at the level of significance (0.05) between mothers whose age is less than 40 years and those 40 years and older in solving problems (t = 2.093, *p* < 0.037) in favor of mothers of the age group 40 years and older (M = 22.00, SD = 1.22), and avoidance (t = 1.987, *p* < 0.048) in favor of mothers under 40 years of age (M = 6.228, SD = 0.464). At the same time, there are no statistically significant differences in social support, re-appraisal of the problem, remorse, positive coping, negative coping, self-efficacy, optimism, hope, resilience, the total score of PsyCap, and PWB due to the variable of the mother’s age (Figure 3).

### 5.2. Prediction of PWB Among Mothers of Autistic Spectrum Children

To assess the predictability of coping strategies and PsyCap of autistic spectrum children’s regarding mothers’ PWB, Multiple Linear Regression Analysis was used, after testing for the presence of multicollinearity. The results are shown in Table 7.

The linear regression results in Table 7, with R^2^ = 0.399, indicate that problem-solving and optimism variables can explain 39.90% of the total changes in PWB of mothers of autistic spectrum children. The ANOVA of the regression model proved to be statistically significant (f = 40.364, *p* < 0.01) and found that problem-solving might influence the changes in the PWB of mothers of autistic spectrum children with a regression coefficient of b = 0.329 (t = 3.697, *p* < 0.01, VIF = 1.198 < 10). Also, the optimism variable might influence the PWB of mothers of autistic spectrum children with a regression coefficient of b = 1.331 (t = 12.600, *p* < 0.01, VIF = 1.169 < 10). In contrast, the avoidance and remorse variables did not statistically significantly predict PWB among mothers of autistic spectrum children (b = −0.288, −0.320, t = 1.050, 1.689 *p* > 0.01, VIF = 1.076, 1.013 < 10), respectively. We can predict the PWB of mothers of autistic spectrum children scores from the following equation:PWB = 13.955 + 1.331 × optimism + 0.329 × problem solving.

## 6. Discussion

The results of the current study show that there were statistically significant differences in avoidance, problem-solving, positive coping, self-efficacy, resilience, PsyCap, and well-being, and the differences were in favor of employed mothers of children with autism spectrum. At the same time, there were no statistically significant differences in social support, re-appraisal of the problem, remorse, negative coping, optimism, and hope. These results are consistent with the study of Shmaisa [62], whose results concluded that the most common strategy for coping with stress among employed mothers is problem-solving. In contrast, social support strategies are the least used. However, these results differ from Abou El-Enein’s [63] study; there were no statistically significant differences in coping with stress according to the mother’s employment status. The work of the mother of autism spectrum children satisfies her psychological needs, such as the need to control things, people, and ideas and make an effort to gain a prestigious position, as well as the need for appreciation, respect, and a sense of security, which increases the level of her PsyCap in terms of resilience and self-efficacy, making her strategies of coping with stress effective and positive. This improves the mother’s PWB and satisfaction with her children’s disability, assisting her in accepting it and enabling her to manage her life affairs with balance.

These results concur with Frech and Damaske [64], whose results indicate that mothers with part-time work with little unemployment have better health at age forty than mothers experiencing persistent unemployment. However, these findings differ from what Keldenick [65] concludedmothers who work part-time, are self-employed, are not employed, and are on maternity leave have higher affective well-being levels than full-time employee mothers. Also, Campos-Serna et al. [66] figured that the double burden role of employed mothers may put additional stress on them, which may negatively affect their mental health outcomes. Also, our results do not concur with Chua [67], found that 29% of employees may faced with well-being challenges through poor mental health. This also contradicts the findings of Al-Shibe’s study [68], which found no statistically significant differences between employed and non-employed women in psychological well-being. This conflicts with the results of previous studies on the differences between employed and non-employed mothers due to the difference in the population in each of these previous studies with the current study, and applied different self-report tools to measure psychological well-being.

These findings regarding the differences between employed and non-employed mothers of children with autism spectrum indicate the need to enhance positive coping strategies, PsyCap, self-efficacy, resilience, and psychological well-being among non-employed mothers of children with autism spectrum.

The results also found no statistically significant differences in coping strategies, PsyCap, and PWB according to the number of children. These results are consistent with the results of [64,65,66,69], which indicated that there were no differences in psychological stress and its response among mothers of children with autism spectrum, according to selected demographic variables, including the number of family members. Al-Shwayat and Al-Sharaa [70] mentioned that the number of members of the autistic family does not affect the psychological stress experienced by the mothers of autistic spectrum children and their coping with it. The impact of the birth of an autistic child on the mother may be more or less the same regardless of the number of family members. The sadness left by the birth of an autistic child includes all mothers regardless of the differences between them in educational qualification and the number of children. The problems found in autistic spectrum children are the same, and the needs related to home care and rehabilitation are the same regardless of the difference between mothers and families.

Mothers of autistic spectrum children, in particular, face many problems because of this stressful experience with the birth of this autistic child, regardless of the number of children in the family, including feelings of disappointment, frustration, rejection, low self-esteem, and distancing from social relationships [71]. This is because the mother is the closest person to her autistic child; she is not only faced with feelings of guilt but also helplessness, and these feelings directly affect her physical and psychological health [72,73]. Perhaps the number of disabled children in the family, whether with autism spectrum disorder or otherwise, is the most essential variable, not the number of non-disabled children in the family, as the number of disabled children in the family is what causes more stress on the parents, especially the mother.

The current study results concluded that there are statistically significant differences in stress coping strategies and PsyCap on the psychological well-being of autistic spectrum children’s mothers. At the same time, there were no statistically significant differences in social support, re-appraisal of the problem, remorse, positive coping, negative coping, self-efficacy, optimism, hope, resilience, the total degree of PsyCap, and PWB due to the variable of the mother’s age. These results are consistent with the study of Jawhar [74] that there are differences in the strategies for coping with stress among mothers with disabilities in favor of the age group over 40 years. The results also agree with what was indicated by [31], which is that factors that affect coping strategies include personal variables such as locus of control, self-esteem, personality type, psychological hardness, and self-efficacy. They also include demographic variables such as age, gender, and economic level.

This result is logical, as older mothers of autistic spectrum children are more experienced in their ability to deal with the problem of their children’s disability and cope with it by solving the problem. In comparison, younger mothers of autistic children lack experience in dealing with their children’s disability and disturbed behaviors, which makes them avoid social interaction. Therefore, older mothers of autistic spectrum children were higher in problem-solving as a coping strategy, and younger mothers were higher in using avoidance as a strategy for coping with the stress of their autistic spectrum child’s disability.

These results indicate that the older mothers of children with ASD use coping strategies focused on solving the problem, dealing directly with the stressful situation, and trying to control it directly by using their resources to solve the problem and confront it through the strategy of searching for a solution of their problems, as well as social support and care through emotional, material [27]. A study conducted by Abdullahi [28] showed that parents of autistic spectrum children use coping strategies through personal resources, support from family and friends, the availability of community resources, and religious strategies to cope with the stress resulting from their negative thoughts and evaluations of themselves and the belief that they cannot change the behavior of their autistic child. In addition, the results indicated that the youngest mothers of a child with autism use emotion-focused coping strategies such as avoidance to cope with the stress resulting from their child’s disability.

The results of the current study also concur with the findings of [75,76,77,78], which indicate that there are no statistically significant differences in PsyCap due to demographic variables. Raising and caring for an autistic child is one of the most pressing problems for the mother. Due to the core characteristics of autism, the child may have meltdowns and display behaviors of concern that may lead to reduced PsyCap and a reduced ability to cope with stress in mothers, regardless of their age, the number of children in the family, and other demographic variables.

The results also indicated that the variables of optimism and problem-solving contributed 39.90% of the total variance in the PWB of autistic spectrum children’s mothers. In this context, the results of Bakhsh’s study [79] showed the existence of a significant relationship between the stressful dimensions of life with anxiety and depression among mothers of disabled children. These results also concur with the study of Al-Zubaidi [75], whose results showed a statistically significant correlation between PWB and stress coping strategies. Hastings et al. [80] and Benson [81] found that optimism is an important predictor of the use of coping strategies among parents with ASD. Also, Willis et al. [82] showed that the parents of children with ASD who have a low level of optimism use avoidance as a coping strategy that exceeds their depressive symptoms.

Lack of optimism and failure to solve problems affect the PWB of mothers of children with autism spectrum. Singhal [76] indicated that the failure of many individuals to face stress and deal with it negatively affects their psychological well-being, such as an increase in the level of symptoms of depression and emotional exhaustion, a decrease in PWB, and psychological burnout; this occurs especially in fathers of persons with disabilities in general. Previous studies [29,41,73] found that the parents of disabled children differ from the parents of normal children in the burdens that affect the PWB of the parents.

The results of several studies and research have confirmed that the birth of a child with a disability or the upbringing of a child with a developmental delay is an event that often has harmful effects on the mental health of parents, who often suffer from increased stress and poor PWB in the absence of effective coping mechanisms and social support [42].

It is essential to cope stress positively, such as by using coping strategies problem-centered and searching for solving it [83]. This study found that dealing with stress and solving problems instead of avoiding them can predict PWB. Many studies have proven the existence of differences in the quality of life, PWB, and happiness, depending on coping strategies, whether negative or positive [79]. Saleh [84] also found that mothers of children who are mentally disabled focused on learning and improving their quality of life.

Thus, solving problems rather than avoiding them and the ability of mothers of children with ASD to rely on their own resources and their PsyCap lead to a high level of PWB and increase their sense happiness and satisfaction with life.

The parents of children with ASD reported many problems in dealing with their child even after diagnosis, in addition to social stigma related to having a child with ASD [85], which may lead to a high level of distress and depressive symptoms compared to parents of non-disabled children [86], therefore we highlighted the importance of problem-solving and optimism-based interventions to promote the PWB of mothers of children with ASD.

## 7. Conclusions

The current study showed that there were statistically significant differences between employed and non-employed mothers of autistic spectrum children in solving problems, avoidance, positive coping, self-efficacy, resilience, PsyCap, and PWB. However, there were no statistically significant differences in social support, problem reassessment, remorse, negative coping, optimism, and hope. Also, there were no statistically significant differences due to the number of children using coping strategies, PsyCap, and PWB. The results also showed there were statistically significant differences between mothers whose age was less than 40 years and those 40 years and older in solving problems in favor of mothers of the age group 40 years and older and avoidance in favor of mothers under 40 years of age. However, there were no statistically significant differences in social support, problem reassessment, remorse, positive coping, negative coping, self-efficacy, optimism, hope, resilience, the total degree of PsyCap, and well-being due to the variable of the mother’s age. Also, the findings found that problem-solving and optimism might influence the PWB of mothers of autistic spectrum children; in contrast, the avoidance and remorse variables did not statistically significantly predict PWB among mothers of autistic spectrum children. These findings revealed the need to develop the ability of autistic spectrum children’s mothers to solve problems and their optimism to enhance their PWB, which may have a positive impact on their autistic spectrum child.

### Limitations and Future Directions

Despite the importance of the current study’s findings, it has several limitations. The current study focused on the mothers of children with autism spectrum without their fathers. Therefore, a similar study can be conducted in the future on the fathers of children on the autism spectrum. Also, this study was confined only to mothers of children with autism spectrum and did not include mothers of children with different disabilities. Thus, future researchers can study the same variables in mothers and fathers of children with different disabilities. In addition, one of the limitations of the current study is the potential bias introduced by self-reported measures, which may impact the results; thus, we recommend that future researchers apply longitudinal studies, experimental designs, or qualitative analyses to explore mothers’ experiences in greater depth. Also, this study used a predictive, descriptive methodology and not an interventional study on the effect of psychological counseling interventions in developing the PWB of mothers of autistic spectrum children; therefore, we need to conduct interventional studies for mothers of autistic spectrum children. In addition, because we did not include the differences in coping strategies, PsyCap, and psychological well-being due to the number of non-disabled children in the family of autistic spectrum children, future researchers can investigate these differences.

## Figures and Tables

**Figure 1 behavsci-14-01235-f001:**
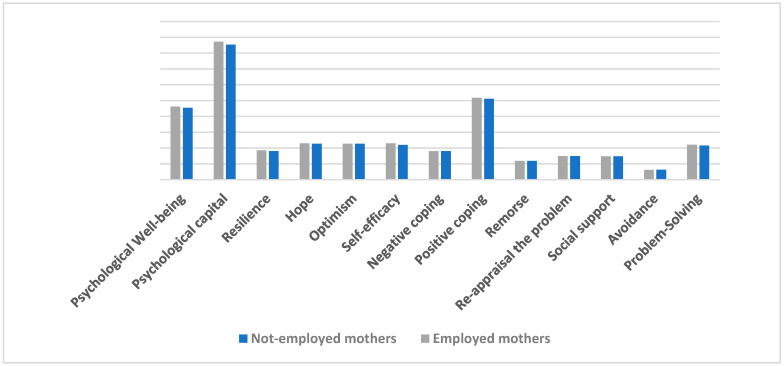
Differences between employed and not employed autistic spectrum children in coping strategies, PsyCap, and PWB.

**Figure 2 behavsci-14-01235-f002:**
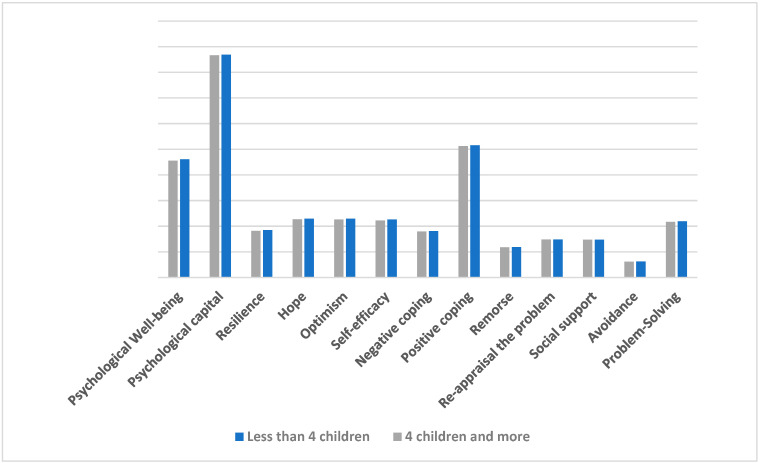
Differences between mothers of autistic spectrum children in coping strategies, PsyCap, and PWB due to the number of children.

**Figure 3 behavsci-14-01235-f003:**
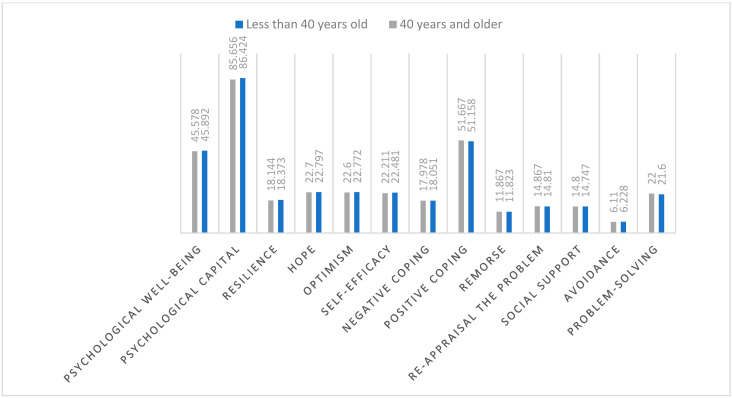
Differences between mothers of autistic spectrum children in coping strategies, PsyCap, and PWB due to the mother’s age.

**Table 1 behavsci-14-01235-t001:** Demographic characteristics of the study participants (*N* = 248).

Variables	*n* (%)
**Mean of Age**	37.613
**Standard Deviation**	4.197
**Age Group**	
Less than 40 years old	158 (63.71%)
40 years and older	90 (36.29%)
**Number of Children**	
Less than 4 children	100 (40.32%)
4 children and more	148(59.68%)
**Work**	
Employed	107 (43.14%)
Not employed	141 (56.86%)

**Table 2 behavsci-14-01235-t002:** Pearson correlation coefficients between items and their dimension total score.

Item	Dimension	Correlation	Item	Dimension	Correlation
1	Problem-solving	0.844 **	15	Social support	0.936 **
2		0.872 **	16		0.879 **
3		0.825 **	17		0.902 **
4		0.806 **	18		0.911 **
5		0.853 **	19		0.895 **
6		0.729 **	20	Re-appraisal of the problem	0.862 **
7		0.835 **	21		0.821 **
8		0.774 **	22		0.883 **
9	Avoidance	0.702 **	23		0.794 **
10		0.782 **	24		0.941 **
11		0.719 **	25	Remorse	0.824 **
12		0.746 **	26		0.851 **
13		0.783 **	27		0.746 **
14		0.791 **	28		0.814 **

** Correlation is significant at the 0.01 level (2-tailed).

**Table 3 behavsci-14-01235-t003:** Pearson correlation coefficients between items and the total scores of their dimension and PCS.

Item	Dimension	R with Dimension	R with PCS	Item	Dimension	R with Dimension	R with PCS
1	Self-efficacy	** 0.828	** 0.779	11	Hope	** 0.809	** 0.770
2		** 0.781	** 0.753	12		** 0.902	** 0.831
3		** 0.825	** 0.733	13		** 0.917	** 0.786
4		** 0.862	** 0.739	14		** 0.926	** 0.829
5		** 0.882	** 0.736	15		** 0.888	** 0.812
6	Optimism	** 0.747	** 0.708	16	Resilience	** 0.894	** 0.837
7		** 0.840	** 0.744	17		** 0.898	** 0.768
8		** 0.776	** 0.747	18		** 0.905	** 0.779
9		** 0.882	** 0.776	19		** 0.853	** 0.785
10		** 0.813	** 0.805				

** Correlation is significant at the 0.01 level (2-tailed).

**Table 4 behavsci-14-01235-t004:** Means, standard deviations, and t-values of differences between employed and not employed mothers of autistic spectrum children.

Measurements	Employed/Not Employed	M	SD	t	Sig. (2-Tailed)
Problem-solving	Not employed	21.496	1.588	3.162	0.002
	Employed	22.075	1.179		
Avoidance	Not employed	6.234	0.487	1.973	0.05
	Employed	6.121	0.381		
Social support	Not employed	14.752	0.965	0.281	0.779
	Employed	14.785	0.869		
Re appraisal of the problem	Not employed	14.823	0.679	0.218	0.828
	Employed	14.841	0.632		
Remorse	Not employed	11.829	0.655	0.256	0.798
	Employed	11.850	0.596		
Positive coping	Not employed	51.071	2.295	2.307	0.022
	Employed	51.701	1.889		
Negative coping	Not employed	18.064	0.864	0.885	0.377
	Employed	17.972	0.733		
Self efficacy	Not employed	21.950	2.185	3.667	0.000
	Employed	22.953	2.062		
Optimism	Not employed	22.688	1.225	0.324	0.746
	Employed	22.738	1.192		
Hope	Not employed	22.624	1.323	1.921	0.056
	Employed	22.944	1.265		
Resilience	Not employed	18.071	1.234	3.338	0.001
	Employed	18.579	1.125		
PsyCap	Not employed	85.333	5.249	2.866	0.005
	Employed	87.215	4.947		
PWB	Not employed	45.447	2.386	2.549	0.011
	Employed	46.215	2.303		

**Table 5 behavsci-14-01235-t005:** Means, standard deviations, and t-values of differences in coping strategies, PsyCap, and PWB due to the variable “number of children” for autistic spectrum children’s mothers.

Measurements	Number of Children Variable	M	SD	t	Sig. (2-Tailed)
Problem-solving	Less than 4 children	21.880	1.335	1.196	0.233
	4 children and more	21.655	1.524		
Avoidance	Less than 4 children	6.230	0.529	1.289	0.198
	4 children and more	6.155	0.382		
Social support	Less than 4 children	14.790	0.769	0.334	0.739
	4 children and more	14.750	1.016		
Re-appraisal of the problem	Less than 4 children	14.860	0.5129	0.577	0.565
	4 children and more	14.811	0.741		
Remorse	Less than 4 children	11.860	0.5129	0.438	0.662
	4 children and more	11.824	0.697		
Positive coping	Less than 4 children	51.530	1.800	1.129	0.260
	4 children and more	51.216	2.352		
Negative coping	Less than 4 children	18.090	0.779	1.053	0.294
	4 children and more	17.979	0.829		
Self-efficacy	Less than 4 children	22.610	2.069	1.346	0.180
	4 children and more	22.229	2.256		
Optimism	Less than 4 children	22.890	1.1625	1.941	0.053
	4 children and more	22.588	1.228		
Hope	Less than 4 children	22.920	1.269	1.570	0.118
	4 children and more	22.655	1.323		
Resilience	Less than 4 children	18.460	1.226	1.820	0.070
	4 children and more	18.176	1.194		
PsyCap	Less than 4 children	86.880	5.036	1.840	0.067
	4 children and more	85.649	5.259		
PWB	Less than 4 children	46.100	2.245	1.760	0.080
	4 children and more	45.561	2.444		

**Table 6 behavsci-14-01235-t006:** Means, standard deviations, and t-values for differences in coping strategies, PsyCap, and PWB due to the mother’s age.

Measurements	Age of Mother	M	SD	t	Sig. (2-Tailed)
Problem-solving	Less than 40 years old	21.601	1.596	2.093	0.037
	40 years and older	22.000	1.122		
Avoidance	Less than 40 years old	6.228	0.464	1.987	0.048
	40 years and older	6.111	0.409		
Social support	Less than 40 years old	14.747	0.951	0.435	0.664
	40 years and older	14.800	0.877		
Re-appraisal of the problem	Less than 40 years old	14.810	0.697	0.650	0.516
	40 years and older	14.867	0.584		
Remorse	Less than 40 years old	11.823	0.653	0.528	0.598
	40 years and older	11.867	0.584		
Positive coping	Less than 40 years old	51.158	2.276	1.800	0.073
	40 years and older	51.667	1.872		
Negative coping	Less than 40 years old	18.051	0.851	0.681	0.497
	40 years and older	17.978	0.734		
Self-efficacy	Less than 40 years old	22.481	2.222	0.935	0.351
	40 years and older	22.211	2.123		
Optimism	Less than 40 years old	22.772	1.215	1.078	0.282
	40 years and older	22.600	1.197		
Hope	Less than 40 years old	22.797	1.339	0.565	0.573
	40 years and older	22.700	1.249		
Resilience	Less than 40 years old	18.373	1.218	1.433	0.153
	40 years and older	18.144	1.195		
PsyCap	Less than 40 years old	86.424	5.275	1.121	0.264
	40 years and older	85.656	5.044		
PWB	Less than 40 years old	45.892	2.398	1.003	0.317
	40 years and older	45.578	2.337		

**Table 7 behavsci-14-01235-t007:** Results of Multiple Regression analysis of coping strategies and PsyCap on PWB of autistic spectrum children’s mothers.

Model	R^2^	Durbin–Watson	F	Independent Variables	Unstandardized Coefficients		Standardized Coefficient	t	Sig.
					B	SE	Beta		
1	0.399	1.184	40.364	Constant	13.955	4.442		3.141	0.002
				Problem-Solving	0.329	0.089	0.201	3.697	0.000
				Avoidance	−0.288	0.274	−0.054	1.050	0.295
				Remorse	−0.320	0.189	−0.085	1.689	0.092
				Optimism	1.331	0.106	0.677	12.600	0.00

## Data Availability

The original contributions presented in this study are included in the article. Further inquiries can be directed to the corresponding author (Boshra A. Arnout).

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
