# Peer review of "The Predictability of Stress Coping Strategies and Psychological Capital on the Psychological Well-Being of Autistic Spectrum Children’s Mothers in the Kingdom of Saudi Arabia"

_behavsci, 2024, doi:10.3390/bs14121235_

Round 1
Reviewer 1 Report
Comments and Suggestions for Authors
This is a very interesting study and one that could add to the existing body of knowledge in the area of autism and familites. However, the language used is utdated and negative. The reviewer has made suggestions for languages changes in the manuscript. Also the discussion could have been more balanced. It would appear from the manuscript that the experience of mothers of autistic children is always negative. It would be good to moderate this by acknowledges some positives but state this will not be discussed in the study
For these reasons revisions are required before publication can be considered.
Abstract:
line 12: We are moving away from a medical model of disability and for this reason the language has changed. This is the suggested language change for here:
"the first breadwinner for her autistic child who has challenges with social communication ..." (APA, 2013)
line14: psychological well-being (PWB)
1. Introduction and Theoretical background
line 45:Suggested change to language: The core characteristics of autism as identified by the DSM 5 (APA, 2013) autistic children have challenges in relation to social communication.
line 81:not sure this is the correct word here. Are you saying mother approach other doctors for a second opinion in the hope that the first doctor's diagnosis in incorrect?
2. Coping with stress among mothers of children with autism spectrum
line 103-106:check that this is on the reference list and why nit is referenced in a different format than other references.
line 123:all behaviors have a communicative intent and therefore it is important that we find out what the child is trying to communicate.
This would allow parents to use pro-active strategies and thus reduce stress for parents and the autistic child. It would be good to include literature in relation to this.
line 138:"for" to have social support
line 104: "treatment"→ support
line 164: "ordinary"→ non-disabled or children without disabilities
3. Purpose of the Present Study
line 183-190:literature
4. Methods
line 242: "working and housewife"→ employed and not employed
7. Discussion
line 311 "showed" → show
line 333:did you check if there was a difference if there was more than on autistic child in the family? If not acknowledge this as a limitation of the study.
line 336:"suffering" →sadness might be a better word here
line342:is this experience traumatic? Is this an assumption that is being made? Has this been checked with mothers/parents?
line 350-353:It is not clear what is meant by this statement.
line 364:coping with it
line 371:indicate
line 382-384:avoid one sentence paragraphs such as this
line 385:“fall in with” concur with the findings
line 387:“bring up"→raising and caring
line388-389:this is very negative. statement. suggested amendment: Due tot he core characteristics of autism the child may have meltdowns and display behaviours of concern.
line 426-427:reassurance and life satisfaction
8. Conclusions
line 429: necessary delete this word
line 438 "was not interested in"→ did not include
line 439 "the other researcher" →future researcher
line 440: see also previous question on mothers with more than one autistic child - another limitation
Author Response
Reply to the comments of the reviewer 1:
This is a very interesting study and one that could add to the existing body of knowledge in the area of autism and familites. However, the language used is utdated and negative. The reviewer has made suggestions for languages changes in the manuscript. Also the discussion could have been more balanced. It would appear from the manuscript that the experience of mothers of autistic children is always negative. It would be good to moderate this by acknowledges some positives but state this will not be discussed in the study
For these reasons revisions are required before publication can be considered.
Abstract:
line 12: We are moving away from a medical model of disability and for this reason the language has changed. This is the suggested language change for here:
"the first breadwinner for her autistic child who has challenges with social communication ..." (APA, 2013)
line14: psychological well-being (PWB)
Reply:
Thank you for your comment, we edited all of these comments in the abstract.
- Introduction and Theoretical background
line 45:Suggested change to language: The core characteristics of autism as identified by the DSM 5 (APA, 2013) autistic children have challenges in relation to social communication.
Reply:
Thank you, we edited it.
line 81:not sure this is the correct word here. Are you saying mother approach other doctors for a second opinion in the hope that the first doctor's diagnosis in incorrect?
Reply:
Yes, we mean that, and we edited it.
- Coping with stress among mothers of children with autism spectrum
line 103-106:check that this is on the reference list and why it is referenced in a different format than other references.
Reply:
Thank you, we went back to reference number 21, thus, reference 21 was cited and not Lazarus, 1966. We edited it
line 123:all behaviors have a communicative intent and therefore it is important that we find out what the child is trying to communicate.
This would allow parents to use pro-active strategies and thus reduce stress for parents and the autistic child. It would be good to include literature in relation to this.
Reply:
Thank you, we edited it.
line 138:"for" to have social support.
Reply:
Thank you, we edited it.
line 104: "treatment"→ support.
Reply:
Thank you, we edited it.
line 164: "ordinary"→ non-disabled or children without disabilities
Reply:
Thank you, we edited it.
Purpose of the Present Study
line 183-190:literature
Reply:
Thank you, we transferred it to the literature section.
- Methods
line 242: "working and housewife"→ employed and not employed
Reply:
Thank you, we edited it.
- Discussion
line 311 "showed" → show
Reply:
Thank you, we edited it.
line 333:did you check if there was a difference if there was more than on autistic child in the family? If not acknowledge this as a limitation of the study.
Reply:
Thank you, we did not investigate the differences according to this variable, thus we added it to the limitations of our study.
line 336:"suffering" →sadness might be a better word here
Reply:
Thank you, we edited it.
line342:is this experience traumatic? Is this an assumption that is being made? Has this been checked with mothers/parents?
Reply:
Thank you, we agreed with you, this stressful experience was not a traumatic experience, we edited it.
line 350-353:It is not clear what is meant by this statement.
Reply:
Thank you, we edited it.
line 364:coping with it.
Reply:
Thank you, we edited it.
line 371:indicate.
Reply:
Thank you, we edited it.
line 382-384:avoid one sentence paragraphs such as this.
Reply:
Thank you, we edited it.
line 385: “fall in with” concur with the findings.
Reply:
Thank you, we edited it.
line 387:“bring up"→raising and caring
Reply:
Thank you, we replaced this word with caring.
line388-389:this is very negative. statement. suggested amendment: Due tot he core characteristics of autism the child may have meltdowns and display behaviours of concern.
Reply:
Thank you, we edited it.
line 426-427:reassurance and life satisfaction
Reply:
Thank you, we corrected it.
Conclusions
line 429: necessary delete this word.
Reply:
Thank you, we deleted it.
line 438 "was not interested in"→ did not include
Reply:
Thank you, we replaced it.
line 439 "the other researcher" →future researcher
Reply:
Thank you, we replaced it.
line 440: see also previous question on mothers with more than one autistic child - another limitation
Reply:
Thank you, we added this limitation to the limitations and future direction section.
Dear reviewer,
we extend our sincere thanks and gratitude to you for your careful and thorough review of our manuscript. We inform you that we have modified and edited all comments, and we hope that the revised manuscript will meet with your acceptance and approval.
Best regards
Authors

Reviewer 2 Report
Comments and Suggestions for Authors
I sincerely thank the journal's editor for the opportunity to review this manuscript. The topic addressed in this article is crucial, as the psychological well-being of mothers of children with autism spectrum disorder (ASD) is of growing interest due to its practical and emotional implications for affected families. However, the article presents numerous shortcomings that require thorough revision and significant restructuring to meet the necessary standards of academic rigor and clarity. Below, I provide a detailed and critical analysis of the areas that need attention.
Introduction
The introduction (lines 41-201) appropriately establishes the relevance of the topic but lacks a clear structure and depth in its literature review. While previous studies are mentioned, the discussion is superficial and does not provide a solid foundation to justify the study's hypotheses.
For example:
- In line 48, it is stated that ASD is "one of the most severe childhood problems," but no epidemiological data or examples are presented to contextualize this claim. I recommend including global or regional statistics on ASD prevalence to reinforce the importance of the topic.
- Concepts such as "coping strategies" and "psychological capital" (lines 97-200) are mentioned without sufficient theoretical review. For instance, the reference to Lazarus (1966) (line 103) is relevant but fails to explore how this framework has developed in recent studies, particularly in contexts similar to Saudi Arabia.
Additionally, clear gaps in the literature are not explored. To justify the study's relevance, the authors must highlight how their research addresses a specific gap beyond simply stating that "little attention has been given to mothers."
Method
The methodological design presents multiple issues that compromise the validity of the results:
- Sample Definition:
- While the sample composition is described (lines 203-209), a detailed explanation of inclusion and exclusion criteria is lacking. For instance, how was it determined that the participants were representative of mothers of children with ASD in Saudi Arabia? Were factors such as socioeconomic status, geographic location, or access to support services considered?
- Instrument Validity and Reliability:
- Although reliability coefficients are provided for the scales used, there is no discussion on whether these tools have been culturally validated for the Saudi context. This is particularly relevant since cultural differences can influence how people respond to questionnaires.
- The "Psychological Capital Scale" and the "Stress Coping Strategies Checklist" are briefly described (lines 211-239), but there is no information on how these tools were adapted or validated specifically for this population. I suggest including details on cross-cultural validation procedures, such as confirmatory factor analysis.
- Statistical Design:
- The regression model used to predict psychological well-being (lines 290-309) seems overly optimistic. An R² of 92.10% is unusually high, suggesting potential overfitting or spurious correlations between variables. This is not addressed in the text, which is a critical omission. The authors should explain how confounding variables were controlled and assess multicollinearity among the independent variables.
- Ethics:
- Although ethical approval is mentioned (lines 455-458), the statement that "no approval was needed due to minimal risks" is questionable. Given that sensitive data on mental health and coping strategies were collected, more detail should be provided on how participants' confidentiality was protected and whether psychological support was offered for potentially distressing cases.
Results
The results section (lines 240-309) presents extensive data but offers limited interpretation and lacks connection to the study's objectives:
- Data Presentation:
- The tables included are not accompanied by clear interpretive descriptions. For instance, Table 2 (lines 247-254) mentions significant differences between working mothers and housewives, but there is no exploration of why these differences are relevant or how they might inform future interventions.
- The graphs (Figures 1-3) lack descriptive labels and are not easily interpretable without additional context in the text.
- Superficial Interpretation:
- In the regression analysis (lines 290-309), although psychological capital and problem-solving are identified as predictors of psychological well-being, the practical implications of this finding are not discussed. How could these results influence psychological support programs or public policies?
- Non-Significant Variables:
- While several factors show no statistical significance, such as the number of children (lines 266-268), the authors do not reflect on why this might be the case or what methodological limitations could have influenced these results.
Discussion
The discussion (lines 310-427) lacks critical reflection on the study's limitations and its relevance within a broader context:
- Connection to the Literature:
- The findings are not adequately connected to previous studies. For example, the assertion that working mothers have higher psychological well-being is not discussed in terms of how this aligns or contrasts with prior research.
- Limitations:
- Although some limitations are mentioned (lines 433-444), they are superficial. For instance, the potential bias introduced by self-reported measures is not addressed, even though it could significantly impact the results.
- Future Directions:
- Future directions are mentioned generically. I suggest that the authors consider longitudinal studies, experimental designs, or qualitative analyses to explore the experiences of mothers in greater depth.
Style and presentation
The article presents numerous formatting and stylistic issues that hinder its readability:
- Writing Style:
- The language is redundant and often unclear. For example, the phrase "mothers are constantly bewildered between their responsibilities" (line 188) is vague and could be reformulated for greater clarity.
- Information is unnecessarily repeated in several sections, increasing the manuscript's length without adding value.
- References:
- The references do not follow a consistent format. For instance, several citations (lines 485-589) lack key details such as DOI or publishers.
In its current state, the article presents significant methodological, conceptual, and stylistic deficiencies that limit its academic contribution. While it addresses a relevant topic, its disorganized structure and superficial analysis prevent the findings from being useful for practice or research. Therefore, I do not recommend its publication in its current format. I suggest that the authors critically revise the article, considering the detailed observations provided in this review. A substantial restructuring, along with more robust analysis and clearer presentation, could significantly improve the manuscript's quality.
Author Response
Reply to the comments of the reviewer 2:
Introduction
The introduction (lines 41-201) appropriately establishes the relevance of the topic but lacks a clear structure and depth in its literature review. While previous studies are mentioned, the discussion is superficial and does not provide a solid foundation to justify the study's hypotheses.
For example:
- In line 48, it is stated that ASD is "one of the most severe childhood problems," but no epidemiological data or examples are presented to contextualize this claim. I recommend including global or regional statistics on ASD prevalence to reinforce the importance of the topic.
Reply:
Thank you, we edited it in the revised manuscript file.
- Concepts such as "coping strategies" and "psychological capital" (lines 97-200) are mentioned without sufficient theoretical review. For instance, the reference to Lazarus (1966) (line 103) is relevant but fails to explore how this framework has developed in recent studies, particularly in contexts similar to Saudi Arabia.
Reply:
Thank you for your comment, we edited it in the revised manuscript file.
Additionally, clear gaps in the literature are not explored. To justify the study's relevance, the authors must highlight how their research addresses a specific gap beyond simply stating that "little attention has been given to mothers."
Reply:
Thank you, we cleared the gaps in reviewing the literature and previous studies at the end of the introduction section, and in the "the purpose of the present study" section.
Method
The methodological design presents multiple issues that compromise the validity of the results:
- Sample Definition:
- While the sample composition is described (lines 203-209), a detailed explanation of inclusion and exclusion criteria is lacking. For instance, how was it determined that the participants were representative of mothers of children with ASD in Saudi Arabia? Were factors such as socioeconomic status, geographic location, or access to support services considered?
Reply:
Thank you, we edited all these comments in the revised manuscript.
- Instrument Validity and Reliability:
- Although reliability coefficients are provided for the scales used, there is no discussion on whether these tools have been culturally validated for the Saudi context. This is particularly relevant since cultural differences can influence how people respond to questionnaires.
- The "Psychological Capital Scale" and the "Stress Coping Strategies Checklist" are briefly described (lines 211-239), but there is no information on how these tools were adapted or validated specifically for this population. I suggest including details on cross-cultural validation procedures, such as confirmatory factor analysis.
Reply:
Thank you for your comment, because of the number of tables, we have briefed the results of the study tools' validity and reliability, but now we have re-wrote it in the revised manuscript file.
- Statistical Design:
- The regression model used to predict psychological well-being (lines 290-309) seems overly optimistic. An R² of 92.10% is unusually high, suggesting potential overfitting or spurious correlations between variables. This is not addressed in the text, which is a critical omission. The authors should explain how confounding variables were controlled and assess multicollinearity among the independent variables.
Reply:
Thank you for your comment, we deleted many variables from the regression model according to the results of the multicollinearity test, and we reperformed the regression analysis.
- Ethics:
- Although ethical approval is mentioned (lines 455-458), the statement that "no approval was needed due to minimal risks" is questionable. Given that sensitive data on mental health and coping strategies were collected, more detail should be provided on how participants' confidentiality was protected and whether psychological support was offered for potentially distressing cases.
Reply:
Thank you for your comment, we obtained the Institutional Review Board of Princess Nourah bint Abdulrahman University (IRB Log Number 23-0495) and we submitted a copy of it to the Journal Editorial Board. The IRB decided to Exempt our study.
Results
The results section (lines 240-309) presents extensive data but offers limited interpretation and lacks connection to the study's objectives:
- Data Presentation:
- The tables included are not accompanied by clear interpretive descriptions. For instance, Table 2 (lines 247-254) mentions significant differences between working mothers and housewives, but there is no exploration of why these differences are relevant or how they might inform future interventions.
- The graphs (Figures 1-3) lack descriptive labels and are not easily interpretable without additional context in the text.
Reply:
Thank you for your comments, we edited all of these in the revised manuscript.
- Superficial Interpretation:
- In the regression analysis (lines 290-309), although psychological capital and problem-solving are identified as predictors of psychological well-being, the practical implications of this finding are not discussed. How could these results influence psychological support programs or public policies?
Reply:
Thank you we edited it after modifying the regression model.
- Non-Significant Variables:
- While several factors show no statistical significance, such as the number of children (lines 266-268), the authors do not reflect on why this might be the case or what methodological limitations could have influenced these results.
Reply:
Thank you for your comment, we edited it in the discussion and limitations and future directions section.
Discussion
The discussion (lines 310-427) lacks critical reflection on the study's limitations and its relevance within a broader context:
- Connection to the Literature:
- The findings are not adequately connected to previous studies. For example, the assertion that working mothers have higher psychological well-being is not discussed in terms of how this aligns or contrasts with prior research.
Reply:
Thank you, we edited it in the revised manuscript.
- Limitations:
- Although some limitations are mentioned (lines 433-444), they are superficial. For instance, the potential bias introduced by self-reported measures is not addressed, even though it could significantly impact the results.
Reply:
Thank you, we added this in the limitations and future direction section.
- Future Directions:
- Future directions are mentioned generically. I suggest that the authors consider longitudinal studies, experimental designs, or qualitative analyses to explore the experiences of mothers in greater depth.
Reply:
Thank you we edited it in the revised manuscript.
Style and presentation
The article presents numerous formatting and stylistic issues that hinder its readability:
- Writing Style:
- The language is redundant and often unclear. For example, the phrase "mothers are constantly bewildered between their responsibilities" (line 188) is vague and could be reformulated for greater clarity.
- Information is unnecessarily repeated in several sections, increasing the manuscript's length without adding value.
Reply:
Thank you, we edited the proofreading for our manuscript.
- References:
- The references do not follow a consistent format. For instance, several citations (lines 485-589) lack key details such as DOI or publishers.
Reply:
Thank you, we edited all references.
In its current state, the article presents significant methodological, conceptual, and stylistic deficiencies that limit its academic contribution. While it addresses a relevant topic, its disorganized structure and superficial analysis prevent the findings from being useful for practice or research. Therefore, I do not recommend its publication in its current format. I suggest that the authors critically revise the article, considering the detailed observations provided in this review. A substantial restructuring, along with more robust analysis and clearer presentation, could significantly improve the manuscript's quality.
Reply:
Dear reviewer
We appreciate your great efforts in reviewing our manuscript, and we have taken all your important comments into consideration, modified all comments in each manuscript section, and highlighted them in the revised manuscript file.
Best regards

Round 2
Reviewer 2 Report
Comments and Suggestions for Authors
I would like to express my sincere gratitude for the revisions made to the manuscript. I believe these adjustments have significantly enhanced the quality and clarity of the article, thoroughly and rigorously addressing the reviewers' comments.
In its current state, I am confident that the article is ready for publication in our journal.
I greatly appreciate your effort and dedication in strengthening this work, which will undoubtedly be a valuable contribution to the field. Please feel free to reach out if there are any additional matters regarding the publication process.